# Eco-evolutionary feedbacks in the human gut microbiome

Benjamin H. Good [1,2,3] ✉ & Layton B. Rosenfeld[4]

Gut microbiota can evolve within their hosts on human-relevant timescales, but little is known about how these changes influence (or are influenced by) the composition of their local community. Here, by combining ecological and evolutionary analyses of a large cohort of human gut metagenomes, we show that the short-term evolution of the microbiota is linked with shifts in its ecological structure. These correlations are not simply explained by expansions of the evolving species, and often involve additional fluctuations in distantly related taxa. We show that similar feedbacks naturally emerge in simple resource competition models, even in the absence of cross-feeding or predation. These results suggest that the structure and function of host microbiota may be shaped by their local evolutionary history, which could have important implications for personalized medicine and microbiome engineering.

The human gut harbors a diverse microbial community comprising hundreds of ecologically interacting strains[1,2]. Recent work has shown that the residents of this community can also evolve over time, through a mixture of within-host evolution[3–9] and the invasion of external strains[4,6,10]. In principle, these rapid genetic changes could alter ecological interactions between species, driving shifts in community composition, and spurring further co-evolutionary responses in other resident strains. Yet despite intensive theoretical speculation[11–16], little is currently known about how the short-term evolution of the microbiota influences (or is influenced by) the composition of its local community.

Previous work has shown that ongoing evolution can alter the ecological composition of small synthetic communities, ranging from simple in vitro systems[17–21] to complex plant-leaf microcosms[22]. Similar effects have also been observed in macro-scale communities with strong trophic structure[23,24]. However, it remains unclear how the feedbacks observed in these tightly coupled settings extend to larger and more metabolically diffuse ecosystems like the gut microbiota. Previous experiments have shown that the disruption of particular metabolic pathways can alter crossfeeding interactions between species of gut bateria in mice and in vitro co-cultures[25,26]. Other variants can enable the colonization of open metabolic niches[27,28]. However, it is

not known how often such ecologically impactful mutations are selected in in situ, or whether the niche partitioning[29] or functional redundancy[1,30] of native gut communities tends to shield them from these evolutionary perturbations.

To distinguish these scenarios, we reanalyzed a large collection of fecal metagenomes from the Human Microbiome Project[1,2], which followed >100 healthy human subjects at 2–3 timepoints over a period of ~6 months (Supplementary Data 1). We hypothesized that this large cohort would provide an opportunity to measure eco-evolutionary feedbacks at a statistical level, by asking whether within-host evolution tends to be accompanied by larger shifts in taxonomic composition during the same time interval. Since gut microbiota also experience daily fluctuations in the absence of evolution[6,31–33], this approach requires us to search for global signals that exceed this baseline variation.

To carry out this analysis, we used a reference-based approach[4] to identify single nucleotide variants (SNVs) that underwent large shifts in frequency within a host between pairs of sequenced timepoints (Methods). These large frequency changes indicate a partial or complete "sweep" within the species in question, in which the focal SNV is likely hitchhiking as a linked passenger mutation[9]. We further classified the sweeps within each species as "evolutionary modification" or

[1]Department of Applied Physics, Stanford University, Stanford, CA 94305, USA. [2]Department of Biology, Stanford University, Stanford, CA 94305, USA. [3]Chan Zuckerberg Biohub—San Francisco, San Francisco, CA 94158, USA. [4]Department of Computer Science, Stanford University, Stanford, CA 94305, USA. ✉e-mail: bhgood@stanford.edu

"strain replacement" events (Supplementary Fig. 1) based on the total number of correlated SNV changes along that species' genome (Methods); we previously showed that these strain replacement events are accompanied by large differences in gene content (-100–1000s of genes), comparable to other circulating strains in the global human population[4]. Using these methods, we compiled a dataset of 16 replacement events and 78 modifications from 799 pairwise genetic comparisons across 45 different species and 134 unique hosts (Supplementary Data 2 and 3). These data provide a unique opportunity to quantify the links between short-term evolution and community structure in the native human gut microbiota in its complex natural environment.

## Results

### Rates of evolution in different community backgrounds

We first asked how the rates of within-species evolution varied with the composition of the local community. We found that the strain replacement and evolutionary modification events were both non-uniformly distributed across taxa. While the sample sizes were too small to resolve the rates of individual species, an omnibus test still revealed a global enrichment of variability across species ($P \approx 0.01$, Methods), with even stronger signals at the family or phylum levels ($P < 10^{-4}$; Fig. 1a, b). Much of this signal was driven by differences between the Bacteroidetes and Firmicutes phyla, with the latter experiencing replacement and modification events at -3-fold higher rates (Fig. 1a, b). Some of these differences are consistent with dispersal-related phenotypes of the resident species[34] (e.g., less strain replacement and more within-host evolution in oxygen- and temperature-sensitive Eubacteriaceae), while others are more mysterious (e.g., higher rates of strain replacement in oxygen-sensitive Ruminococcaceae).

We also investigated how the strain replacement and evolutionary modification events depended on the global properties of their surrounding community. For example, classical theories suggest that the rates of evolution may strongly depend on local species diversity[35,36], either by closing off environmental niches[37–39], or by creating new opportunities to adapt to metabolic byproducts[18,40,41] or other interactions[42] with the surrounding community members. Across our cohort, we observed a weak positive correlation between the rate of evolutionary modification and the Shannon diversity at the initial timepoint, after controlling for differences between phyla ($P \approx 0.04$, logistic regression; Supplementary Table 1; Fig. 1c). This shows that community diversity does not strongly limit the rate of evolution over the range of diversity found in healthy human gut microbiomes.

The weak positive correlation in Fig. 1c could be consistent with some models of accelerated community evolution [see ref. 43 for a related analysis]. However, the effect size associated with the community diversity was much smaller than the differences in the modification rate between phyla. Moreover, this signal was primarily driven by the top diversity decile in the Bacteroidetes phylum, rather than a systematic trend across all samples ($P \approx 0.2$ when the top decile of the diversity distribution is excluded; Supplementary Table 1). We also observed no correlation between the initial community diversity and the rate of strain replacement ($P \approx 0.8$; Supplementary Table 1, Supplementary Fig. 3). Consistent with these observations, we found that there were no strong correlations between the replacement and modification events in different species in the same host (Fig. 1d), as might be expected if genetic changes were primarily determined by a global property of the community like taxonomic diversity (Fig. 1c and Supplementary Fig. 2) or the Firmicutes-to-Bacteroidetes ratio (Supplementary Fig. 4). These different lines of evidence suggest that to a

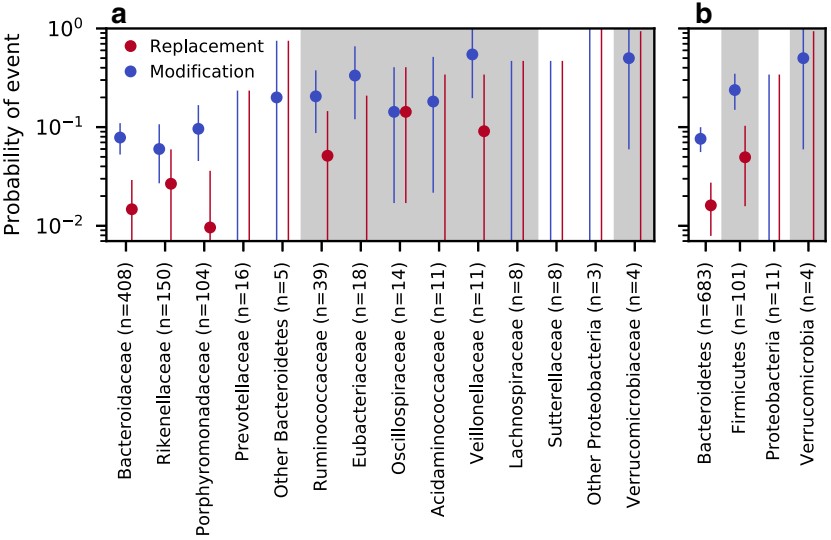

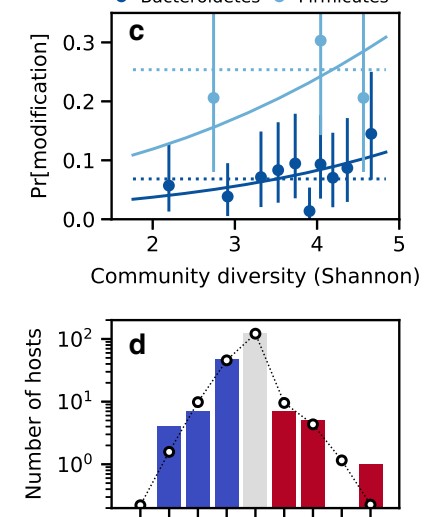

**Fig. 1 | Replacement and modification rates vary across species, but only weakly depend on the diversity of the surrounding community.** Fraction of resident populations in which we detected strain replacement or evolutionary modification events (Methods), coarse-grained at the family (**a**) or phylum level (**b**). Lines denote 2.5–97.5 percentiles of the Gamma posterior distribution obtained from the observed number of counts (Methods). **c** The probability of a modification event as a function of the species diversity of the surrounding community. Points show the fraction of modification events in Bacteroidetes or Firmicutes species stratified by quantiles of the Shannon diversity; vertical lines denote 2.5–97.5 percentiles of the Gamma posterior distribution that were computed as above. Solid lines illustrate the best-fit logistic regression model using the phylum and community diversity as predictor variables (Methods), while the dashed lines show the average of each phylum computed for the bottom 90% of community diversity values. The analogous regression for strain replacements events was not statistically significant ($P \approx 0.8$; Supplementary Fig. 3). **d** The distribution of the number of genetic changes per community. Bars denote the fraction of community comparisons with a given number of genetic events (replacements + modifications). These comparisons were further partitioned into two subgroups: those that only experienced modification events (left) and those that also experienced one or more replacement events (right). Lines denote the null distribution obtained by randomly permuting replacement and modification events across resident populations of the same species (Methods).

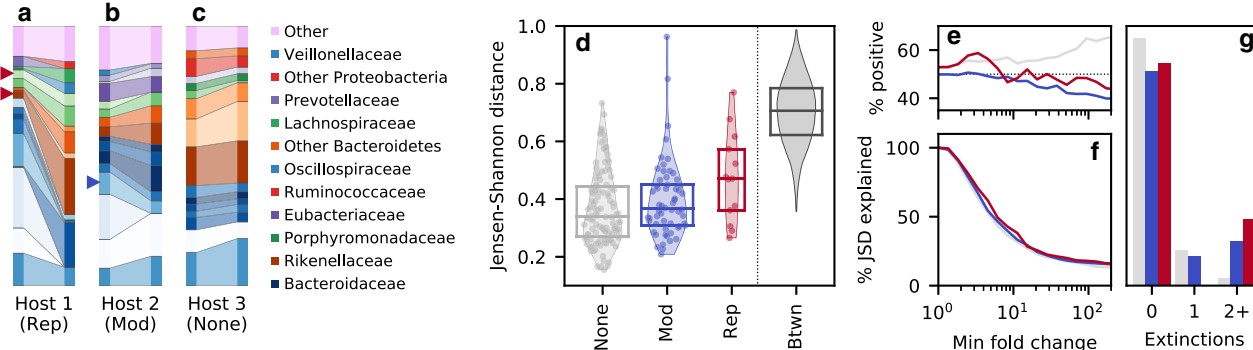

**Fig. 2 | Genetic changes within species are statistically associated with shifts in taxonomic composition over time.** Species relative abundances over time in three example hosts experiencing strain replacement events (**a**), evolutionary modification events (**b**), and no genetic changes (**c**). Species are shown if they had a relative abundance > 2% in one of the hosts in at least one timepoint; all other species are grouped together in the "Other" category. Species experiencing replacement and modification events are indicated by the triangles. **d** Distribution of Jensen-Shannon distances over time (Methods) for communities that experienced at least one replacement (Rep, n = 13) or modification event (Mod, n = 58), or no genetic changes (None, n = 123). Symbols denote individual data points, while boxes show the median and inter-quartile range; the JS distances between ~1000 random pairs of hosts are shown at right for comparison. **e**, **f** For each of the communities in (**d**), the fraction of the Jensen-Shannon divergence explained by fold changes greater than a given amount (**f**, Methods), as well as the fraction of this amount explained by positive vs negative changes (**e**). Colors indicate the Rep, Mod, and None categories in (**d**). **g** The number of abundant species that went extinct in each of the communities in (**d**) (Methods). Colors are the same as (**e**).

first approximation, the genetic changes within our cohort can be described by a simple null model, where replacement and modification events occur approximately independently in different resident populations, but with species-specific rates.

## Genetic changes are correlated with shifts in composition

Armed with this information, we next asked how the genetic changes within species were correlated with shifts in their ecological structure over time. Within our cohort, one can find individual hosts with replacement or modification events that were also accompanied by large changes in species composition between the same two timepoints (Fig. 2a, b; Supplementary Data 4). One can also find hosts with no detected genetic changes that experienced smaller ecological shifts (Fig. 2c). To quantify this pattern more systematically, we partitioned our 194 community comparisons into three categories depending on whether the community had at least one replacement event, at least one modification event, or no genetic changes within species (Fig. 2d). For each of these communities, we also calculated the Jensen-Shannon (JS) distance between the species abundance distributions at the initial and final timepoints (Methods); this provides a simple metric for quantifying the overall change in species-level composition over time[44].

While species abundances can fluctuate due to a variety of intrinsic and extrinsic factors[6,31–33], we observed a small but systematic trend toward larger JS distances in hosts that experienced an evolutionary modification event, and even larger distances in hosts that experienced a strain replacement event (Fig. 2d). To assess the significance of these trends, we utilized an empirical null model informed by Fig. 1: we generated $n = 10^4$ bootstrapped datasets by randomly permuting the observed genetic events within the resident populations of each species. By construction, these bootstrapped datasets preserve both the overall number of genetic changes of each type, their distribution across species, and the species abundance trajectories within each host. This provides a principled way to test for associations between ecological and genetic changes, while preserving the complex correlations between the abundances of different species in large microbial communities.

We found that both replacement and modification events had significantly larger JS distances than expected under this empirical null model ($P \approx 0.001$ and $P \approx 0.003$, respectively; Methods), suggesting that the ecological and genetic changes within these communities are indeed correlated with each other. Notably, the evolutionary

modification signal was primarily driven by a depletion of the smallest temporal fluctuations (Supplementary Fig. 6), rather than the handful of hosts with the largest ecological shifts. By contrast, the strain replacement signal was driven by a global increase in JS distances (Supplementary Fig. 6), which could be consistent with both an increased power to detect these genome-wide events, as well as a larger phenotypic impact due to the larger number of SNV and gene-content differences that they carry. Similar differences were observed when community composition was measured at the genus, family, or phylum levels (Supplementary Fig. 5). We also observed a small but significant enrichment of "extinction" events in abundant species (modifications $P \approx 0.003$, replacements $P \approx 0.01$; Methods) (Fig. 2g), which suggests that these eco-evolutionary correlations are not specific to the idiosyncratic features of the Jensen-Shannon metric.

The ecological distances in Fig. 2d can be further decomposed into the contributions from different species (Methods). Interestingly, this decomposition shows that the larger distances in Fig. 2 were not solely driven by expansions of the focal species (i.e., those with genetic changes), as expected under some of the simplest models of niche expansion[22,37]. While we observed a small bias toward expansions in species that experienced a modification event ($P \approx 0.05$, Methods), the focal species still declined in frequency in more than 40% of cases (Fig. 3a, b). Strain replacements displayed a similar trend: while the focal species experienced significantly larger fold-changes over time ($P \approx 0.03$, Methods), the majority of these changes involved declines in relative abundance rather than expansions (Fig. 3a, b). Furthermore, we observed no strong correlation between the community-level distances in Fig. 2d and the fold changes in the focal species (Fig. 3d). This suggests that the eco-evolutionary correlations in Fig. 2d do not depend on the direction of the focal species's trajectory. Rather, the observed ecological shifts are more global in nature, in that they are comprised of correlated shifts in the abundances of other species in the community, even when focal species itself declines.

Figure 2b illustrates a prototypical example of this behavior: we detected an evolutionary modification event in one species (*Bacteroides stercoris*), which declined slightly in relative abundance, while two other species went extinct in the same time interval—one from the same genus (*Bacteroides massiliensis*) and another from a different bacterial family (*Barnesiella intestinihominis*). To quantify the relationships between these species more systematically, we calculated the fraction of JSD that was contributed by species in the same family as one of the focal species (Fig. 3f). We found that the family-level

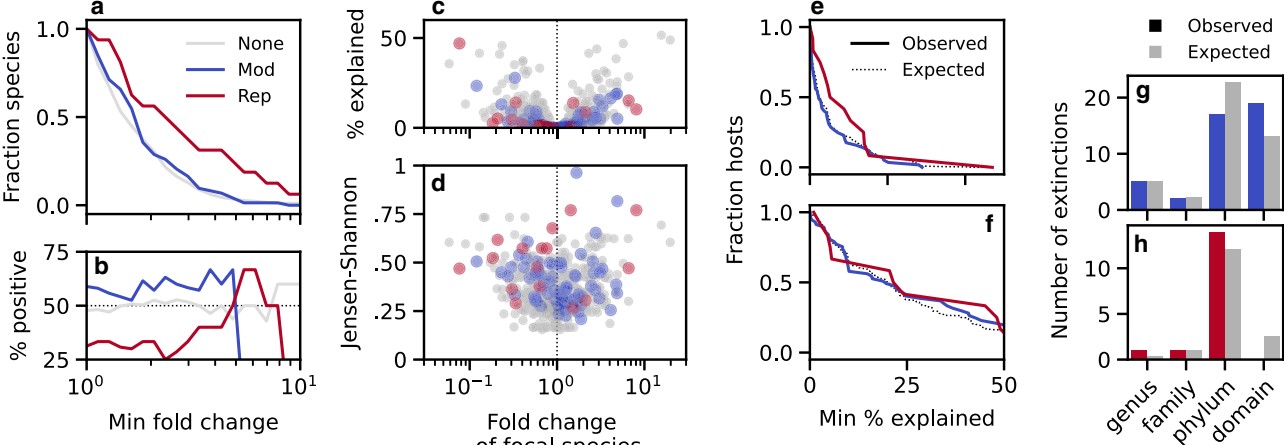

**Fig. 3 | Ecological changes are not solely explained by frequency increases in focal species. a, b** Fraction of focal species with absolute fold changes in relative abundance greater than a given amount, regardless of direction (**a**), and the fraction of this amount contributed by positive vs negative changes (**b**). **c, d** The Jensen-Shannon distances in Fig. 2d as a function of the fold change in relative abundance of the focal species (**d**), and the fraction of the Jensen-Shannon divergence explained by each of these fold changes (**c**, Methods). **e** The fraction of community comparisons in which the Jensen-Shannon divergence explained by the focal species exceeds a given percentage. Solid lines denote the observed data for the replacement (red) and modification (blue) hosts in Fig. 2d, while the dashed lines denote the expectations of a null model where the focal species are chosen at random (Methods). **f** An analogous version of panel e using the fraction of Jensen-Shannon divergence explained by all species in the same family as one of the focal species. **g, h** Taxonomic relationship with the most closely related focal species for each of the extinction events in Fig. 2g. Colored bars denote the observed data for the modification (top) and replacement (bottom) hosts, respectively, while the gray bars denote the corresponding expectations from the null model in (**e**).

contributions were somewhat larger than the contributions of the focal species themselves (Fig. 3f), consistent with the larger number of species involved, but they were not significantly different than expected by chance ($P \approx 0.07$; Methods). Similar results were obtained for other taxonomic groupings (Supplementary Fig. 9), and for the species that went extinct between the two timepoints (Fig. 3g, h). This shows that the statistical signal in Fig. 2 is not solely driven by fluctuations in closely-related species, but rather by correlated fluctuations in distantly related taxa.

### Similar feedbacks emerge in resource competition models

Our results establish a statistical link between the short-term evolution of the human gut microbiota and changes in its ecological structure. Such correlations could arise through several causal scenarios: genetic changes could alter ecological interactions between species, driving shifts in community composition ("evolution-driven feedbacks"); alternatively, environmental perturbations could lead to taxonomic shifts while also creating new opportunities for within-host evolution ("ecology-driven feedbacks"). It is difficult to distinguish these scenarios from observational data alone, though the independent distribution of genetic events (Fig. 1d) and the slight bias toward the expansion of species with evolutionary modification events ($P \approx 0.05$, Methods; Fig. 3b, d) could provide some evidence in favor of the evolution-driven scenario.

To gain more intuition for the range of possible behaviors, we also studied the correlations between ecological and genetic changes in a simple mathematical model, where the contributions of these two feedback mechanisms could be precisely controlled. We considered a simple class of resource competition models[15,45–49], in which $\mathcal{S}$ coexisting species compete for $\mathcal{R}$ substitutable resources that are continuously supplied by the environment (Fig. 4a; Supplementary Note 1). For a saturated community ($\mathcal{S} = \mathcal{R}$) at ecological equilibrium, previous work has shown that the selection pressures on new mutations are independent of the external environment and the abundances of the resident species (ref. 15; Supplementary Note 1). This constitutes a simple theoretical example in which ecology-driven feedbacks are effectively damped by the collective response of the community. In this saturated case, one can derive a further result

connecting the invasion fitness of a small-effect mutation ($S_{inv}$) with the ecological perturbations that it causes after it invades (Fig. 4b). This relationship can be written in the implicit form,

$$S_{inv} = C_{\mu^*} \cdot \left( \Delta X_{\mu^*} \cdot f_{\mu^*} + \sum_{\mu=1}^{\mathcal{S}} \Delta f_\mu \cdot X_\mu \right), \quad (1)$$

where $f_\mu$ denotes the relative abundance of species $\mu$, $X_\mu$ is the logarithm of its total resource uptake rate, $\mu^*$ denotes the focal species in which the mutation occurred, and $C_{\mu^*}$ is a proportionality constant that depends on the focal species $\mu^*$ (and its interactions with the surrounding community) but is otherwise independent of the phenotypic impact of the mutation (Supplementary Note 1).

A special case of Eq. (1) occurs when a mutation increases the total resource uptake rate of the focal species ($\Delta X_{\mu^*} > 0$) but leaves its relative uptake rates intact. In this case, a successful mutation will always increase the relative abundance of the focal species ($\Delta f_{\mu^*} > 0$), while causing downstream shifts in other species that depend on the metabolic overlap within the larger community (ref. 15; Supplementary Note 1). On the other hand, when mutations are able to alter the relative uptake rates of the focal species, Eq. (1) shows that any combination of directions of $\Delta f_\mu$ are possible (at least in principle), provided that they lead to the same overall sign on the right-hand side.

This effect is present even for the simplest case of two resources, where a graphical picture provides some further intuition about the underlying mechanisms involved (Fig. 4c). Previous work has shown that differences in the overall uptake rates of the two species ($X_\mu$) create local selection pressures for mutations that shift metabolic effort toward the species with the smaller value of $X_\mu$ (ref. 15; white region in Fig. 4c). Mutations in this less fit species will be favored if they drive the relative uptake rates further away from the fitter competitor; these mutations will generally lower the abundance of the focal species ($\Delta f_{\mu^*} < 0$), unless they are compensated by a corresponding increase in $X_{\mu^*}$.

This shows that natural selection can favor mutations that lower the abundance of the focal species ($\Delta f_{\mu^*} < 0$), which could provide a potential mechanism for the empirical behavior observed in Fig. 3b. It also shows that mutations can produce a mixture of positive and

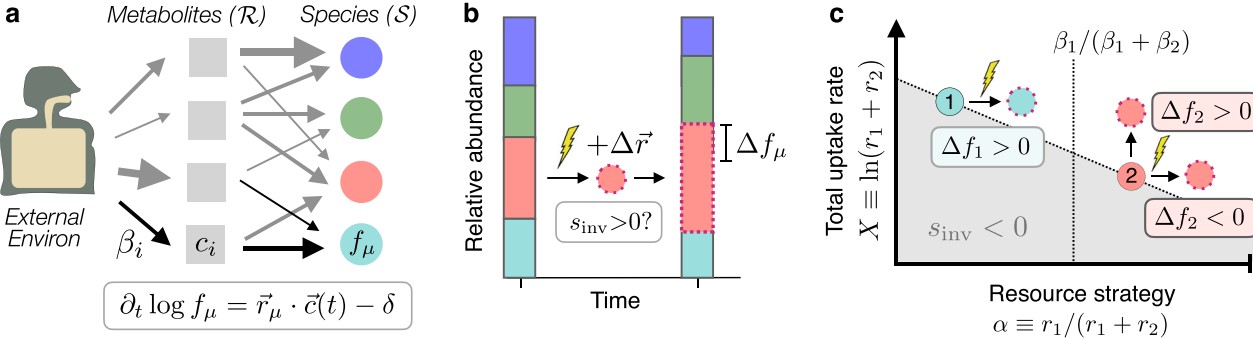

**Fig. 4 | Eco-evolutionary feedbacks in a simple resource competition model.**
**a** A simplified model of a microbiome: $\mathcal{S}$ resident species compete for $\mathcal{R}$ substitutable metabolites that are continuously supplied by the environment; $r_{\mu,i}$ denotes the uptake rate of metabolite $i$ by species $\mu$, while $\beta_i$ denotes the input flux from the environment. In the absence of further mutations, growth and dilution lead to steady-state levels of resource concentrations ($c_i$) and species abundances ($f_\mu$). **b** An example of an evolution-driven feedback: a mutation in a focal strain alters its resource uptake phenotype by an small amount $\Delta\vec{r}$; if the mutation provides a fitness benefit ($S_{inv} > 0$), its descendants will proliferate in the community and cause a shift in the steady-state abundances of the resident species

($f_\mu \to f_\mu + \Delta f_\mu$). Colors denote the different species in (**a**). **c** Evolution-driven feedbacks in a simplified community with $\mathcal{S} = \mathcal{R} = 2$ (see Supplementary Note 1 for a mathematical derivation). Solid circles denote the resident strains, while dashed circles denote potential mutants. Shading indicates regions of phenotype space that are favored ($S_{inv} > 0$, white) or disfavored ($S_{inv} < 0$, shaded) to invade; when $\mathcal{S} = \mathcal{R} = 2$, these regions are completely determined by the resident strains, and are conditionally independent of the input fluxes from the external environment ($\beta_i$). The top two mutations lead to an increased abundance of the focal species when they invade, while the bottom example shows a beneficial mutation ($S_{inv} > 0$) that decreases the relative abundance of its focal species ($\Delta f_{\mu'} < 0$).

negative abundance changes across a large range of species, regardless of their phylogenetic or phenotypic similarity.

These theoretical results show that simple evolution-driven feedbacks can recapitulate some of the qualitative features of Figs. 2 and 3, even in the absence of additional factors like crossfeeding[50,51], spatial structure[52], or phage predation[53]. Nevertheless, this analysis also has important limitations: Eq. (1) only applies to the simplest case of small-effect mutations in ecologically saturated communities. Further theoretical work is necessary to understand how these results extend to the non-saturated case[49], and to incorporate important factors like cross-feeding[50,51] and spatial structure within the gut[52]. Perhaps more importantly, Eq. (1) only shows that it is theoretically possible for a beneficial mutation to produce a given combination of $\Delta f_\mu$ values, provided that it changes the resource uptake phenotype of the focal strain in a particular way. The extent to which this will occur in practice will depend on how easily the corresponding changes can be produced by mutations in the resident strains. This requires additional information about the genetic architectures of the resource uptake phenotypes, which are poorly constrained by existing data. Our results suggest that it would be interesting to map out the accessible ecological perturbations in future experiments, e.g., by measuring the joint distribution of $S_{inv}$ and $\{\Delta f_\mu\}$ values obtained by mutating or swapping resident strains. This joint distribution would constitute an eco-evolutionary analog of the distribution of fitness effects of new mutations, which plays a central role in evolutionary genetics[54].

## Discussion

Further work will be required to determine which of the various ecological or evolutionary feedback mechanisms are most relevant for the human gut, and to understand the mix of external (or internal) factors that drive them. Our results show that the statistical signature of these feedbacks can sometimes be observed even in complex in situ environments, generating targeted hypotheses that could be tested in future experiments (e.g., using large synthetic gut communities[55,56]). These results could have important implications for the development of personalized therapies that aim to tune the composition of the gut microbiota. For example, within-host evolution could perturb the ecological structure of the community away from its desired state, requiring repeated interventions in order to control. Conversely, shifts in the local environment could select for different sets of mutations in

different hosts, which could potentially ripple through the community in unpredictable ways. The phylum-specific rates of evolutionary modification and strain replacement we observed in Fig. 1 could be interesting in this context, since the Firmicutes-to-Bacteroidetes ratio has previously been associated with several markers of host health and disease[57]. Further work will be required to determine whether within-host evolution could contribute to these associations.

While we have focused on applications to the human gut microbiota here, our statistical framework can also be applied to other microbial ecosystems where large cohorts of independent evolutionary replicates are available. Extending our pairwise approach to denser longitudinal timeseries[6,9,58] could help disentangle the relative contributions of evolution- vs ecology-driven feedbacks. These and other generalizations will be useful for quantifying the links between short-term evolution and community structure in other microbial ecosystems characterized by large numbers of coexisting species[39,58–60].

## Methods

### Metagenomic pipeline

We utilized a collection of shotgun metagenomic data from the Human Microbiome Project[1,2] that we analyzed in a previous study[4]. This cohort contained a total of 316 fecal samples from 138 healthy human subjects, who were sequenced at 2–3 timepoints roughly 6 months apart (Supplementary Data 1). We chose to focus on this cohort for several reasons: (i) the samples were sequenced with relatively high coverage, which enhances our ability to detect evolutionary changes within species; (ii) the subjects were specifically selected to avoid unusual external perturbations (e.g., antibiotics, chronic diseases, or large dietary shifts) that could have an outsized influence on microbiome composition; and (iii) we previously showed[4] that the sampling interval in this cohort results in a mixture of hosts with genetic changes in some of their resident species, and other hosts with no genetic changes; this mixture of outcomes is critical for our present analysis.

We processed these data using the pipeline described in ref. 4. Briefly, the MIDAS software package[61] was used to align the raw sequencing reads from each sample to a large panel of reference genomes representing different species. The relative abundance of each species was estimated in each sample based on the coverage of single copy marker genes (Supplementary Data 4). These relative

abundances were used to assemble a personalized panel of reference genomes for each host, which were used to identify single nucleotide variants within each species. These initial SNVs were subsequently filtered based on their absolute and relative coverage, as well as their location along the genome, using the default parameters and code provided in ref. 4.

## Detecting replacement and modification events within species

We used these data to detect replacement and modification events between pairs of timepoints using the methods described in Ref. 4. For each pair of time points, we identified the subset of "quasi-phaseable" species where the lineage structure was sufficiently simple that the dominant lineage could be identified with a high degree of confidence[4]. Within each of these quasi-phaseable populations, we calculated the total number of SNVs that transitioned from <20% frequency in one timepoint to >80% frequency in another (or vice versa), along with the corresponding number of changes expected from sequencing noise alone[4]. As in ref. 4, we recorded zero SNV differences if the estimated false positive rate was >10%. The resulting numbers and locations of the SNV differences in each of the populations are listed in Supplementary Data 2 and 3. These counts were used to classify each population as experiencing a replacement event (>100 SNV differences), a modification event (<100 SNV differences), or zero genetic changes; we previously showed that this divergence-based definition is consistent with the patterns of private marker SNV sharing on these time scales[4] (Supplementary Fig. 1). In carrying out this analysis, we noticed that the followup samples from two subjects (Subject IDs 763536994 and 763880905) were possibly swapped, since they exhibited strain replacement events in all of their resident species, but had nearly identical strains when compared to the corresponding sample from the other host. For simplicity, we omitted these two subjects from all of our downstream analyses (though they are still included in Supplementary Data 2 and 4 for completeness). This analysis yielded a total of 18 replacement events and 98 modification events from a total of 937 pairwise comparisons across 45 different species in 136 unique hosts.

Since some hosts were sampled at more than two timepoints, we further de-replicated these data to ensure that all genetic comparisons were performed on non-overlapping time intervals. For each host-species combination, we only included pairwise comparisons from consecutive quasi-phaseable timepoints; all other pairwise comparisons were treated as missing data (similar to non-quasi-phaseable timepoints). This de-replication procedure yielded a final dataset containing 16 replacement events and 78 modification events from a total of 799 pairwise comparisons across 45 different species in 134 unique hosts. These data were used for all of our subsequent analyses.

## Quantifying heterogeneity in replacement and modification rates across species

We quantified the overall variability in the rates of replacement and modification events across species using a global likelihood ratio test,

$$\Lambda \equiv \sum_{\mu,e} n_{\mu,e} \log\left(\frac{p_{\mu,e}}{\bar{p}_e}\right), \tag{2.a}$$

where $n_{\mu,e}$ is the total number of events of type $e \in \{R, M, 0\}$ in species $\mu$ (representing replacements, modifications, and no genetic changes, respectively), and

$$p_{\mu,e} = \frac{n_{\mu,e}}{\sum_{e'} n_{\mu,e'}}, \qquad \bar{p}_e = \frac{\sum_\mu n_{\mu,e}}{\sum_{\mu,e'} n_{\mu,e'}}. \tag{2.b}$$

Equation (2.a) quantifies the degree to which the observed replacement and modification rates deviate from a null model in which these events occur uniformly across species. We assessed the significance of

this deviation by comparing the observed value of $\Lambda$ to a null model in which the replacement and modification events were randomly permuted across all quasi-phaseable samples. This defines a corresponding $P$-value,

$$P = \Pr\left[\Lambda \geq \Lambda^{obs}\right], \tag{3}$$

which we estimated numerically from $n = 10^4$ bootstrapped samples (Supplementary Code 1). We performed this test for the observed species counts $n_{\mu,e}$, as well as coarse-grained versions that merged the observed counts at the genus, family, or phylum levels (Fig. 1a, b).

The confidence intervals in Fig. 1 were obtained from the $\alpha/2$ and $1 - \alpha/2$ percentiles of the posterior distribution of the underlying Poisson process,

$$p(p_{\mu,e} | \vec{n}) \propto p_{\mu,e}^{n_{\mu,e}-1} e^{-p_{\mu,e}\sum_e n_{\mu,e}}, \tag{4}$$

with $\alpha = 0.05$. In the case where $n_{\mu,e} = 0$, the posterior is an improper distribution, so we define the lower limit of the confidence interval to be 0, and the upper limit to be the point where $e^{-p_{\mu,e}\sum_e n_{\mu,e}} \sim \alpha/2$.

## Quantifying the relationship between community diversity and the rates of evolutionary modification

We quantified the residual effects of community diversity on the rates of evolutionary modification using a logistic regression model, in which the probability of an evolutionary modification in species $\mu$ in community $c$ is given by

$$\log\left(\frac{p_{\mu,c}}{1 - p_{\mu,c}}\right) = \beta_0 + \vec{\beta}_P \cdot \vec{P}_\mu + \beta_H H_c^0, \tag{5.a}$$

where $\vec{P}_\mu$ is an indicator variable giving the phylum of species $\mu$, and $H_c^0$ is the Shannon entropy of the surrounding community,

$$H_c^0 = - \sum_\nu f_{\nu,0} \log_2 f_{\nu,0}, \tag{5.b}$$

which was estimated from the relative abundances of the species at the initial timepoint. For simplicity, we only examined species in the Bacteroidetes and Firmicutes phyla, which constitute the vast majority of our data (Fig. 1b). Regression coefficients and $P$-values were estimated using the logistic regression routines in the `statsmodels` library[62] using default parameter values. Similar results were obtained when $H_c^0$ was replaced with a measure of species richness (Supplementary Fig. 2; Supplementary Table 1).

## Quantifying the overdispersion of genetic changes within hosts

We quantified the residual effects of more general community properties on replacement and modification rates by examining the overdispersion of genetic changes within a given host community. For each pair of timepoints, we calculated the total number of quasi-phaseable populations that experienced a replacement or modification event, and examined the distribution of these counts across our dataset (Fig. 1d). We compared this distribution to a null model in which the replacement and modification events were randomly permuted across quasi-phaseable populations of the same species. This permutation scheme eliminates any correlations between the genetic changes in different resident populations, but preserves the heterogeneity in the number and types of quasi-phaseable populations in different hosts, and the total number of replacement and modification events in different species.

## Quantifying shifts in ecological structure over time

We quantified the changes in ecological structure between a pair of timepoints using the Jensen-Shannon divergence,

$$JSD = \sum_{\mu} \left[ f_{\mu,0} \log_2 \left( \frac{2f_{\mu,0}}{f_{\mu,0}+f_{\mu,1}} \right) + f_{\mu,1} \log_2 \left( \frac{2f_{\mu,1}}{f_{\mu,0}+f_{\mu,1}} \right) \right], \quad (6.a)$$

$$= \sum_{\mu} \bar{f}_{\mu} \left[ \left( \frac{2}{1+r_{\mu}} \right) \log_2 \left( \frac{2}{1+r_{\mu}} \right) + \left( \frac{2}{1+r_{\mu}^{-1}} \right) \log_2 \left( \frac{2}{1+r_{\mu}^{-1}} \right) \right], \quad (6.b)$$

where $f_{\mu,0}$ and $f_{\mu,1}$ are the relative abundances of species $\mu$ at the initial and final timepoints, $\bar{f}_{\mu} = (f_{\mu,0}+f_{\mu,1})/2$ is the average relative abundance, and $r_{\mu} = f_{\mu,1}/f_{\mu,0}$ is the fold change between the two time points. The Jensen-Shannon distance (Fig. 2d) is defined as the square root of the Jensen-Shannon divergence ($d = \sqrt{JSD}$).

We quantified the differences between the distributions of Jensen-Shannon distances in Fig. 2d using a one-sided Kolmogorov-Smirnov (KS) test,

$$D_e = \max_d \left\{ \hat{S}_e(d) - \hat{S}_0(d) \right\}, \quad (7)$$

where $\hat{S}_e(d)$ is the fraction of communities of type $e \in \{R, M, 0\}$ with Jensen-Shannon distance larger than $d$. Larger values of $D_e$ indicate an enrichment of larger Jensen-Shannon distances, relative to communities with no genetic changes. We assessed the significance of these differences by comparing the observed $D_e$ values to a null model similar to Fig. 1d, in which the replacement and modification events were randomly permuted across quasi-phaseable populations of the same species. As mentioned above, this permutation scheme preserves the heterogeneous opportunities for replacement and modification events in different hosts, the observed distribution of Jensen-Shannon distances, and any correlations between the two. This defines a corresponding $P$ value,

$$P = \Pr \left[ D_e \geq D_e^{obs} \right], \quad (8)$$

which we estimated numerically from $n = 10^4$ bootstrapped samples. For completeness, we used a similar approach to compare the overall change in Shannon diversity between the two timepoints (Supplementary Fig. 7). We observed a much weaker correlation with the number of replacement or modification events, suggesting that the signals in Fig. 2 cannot be explained by diversity fluctuations alone.

In addition to these entropy-based metrics, we also quantified shifts in community structure by examining extinction events among highly abundant species (Fig. 2g). In this calculation, a species was counted as going extinct if it transitioned from an initial relative abundance > 1% to a final relative abundance < 0.01%. We compared the distributions of extinction events in Fig. 2g using a similar KS test as Fig. 2d, with the number of extinction events in a given community (Supplementary Fig. 8) replacing the Jensen-Shannon distance in Eq. (7); we assessed the significance of these differences using the same permutation-based null model as above. We also compared these observations to the corresponding number of "invasion" events, defined as the time-reversed version of an extinction event above (Supplementary Fig. 8). We quantified the asymmetry between extinctions and invasions by comparing them to a null model in which we randomly flipped the direction of time in each sample.

We used a similar approach to compare the distributions of focal species fold changes Fig. 3b, with the magnitude of the log fold change, $|\log(f_{\mu,1}/f_{\mu,0})|$, replacing the Jensen-Shannon distance in Eq. (7). We quantified the excess of positive over negative changes (Fig. 3a) using a one-sided sign test. Similar results were obtained by comparing the fraction of positive changes in species with modifications vs no genetic changes using the permutation-based null model above ($P \approx 0.02$).

## Decomposing the Jensen-Shannon divergence

The linear sum in Eq. (6.b) provides a natural way to decompose the Jensen-Shannon divergence into contributions from different species. For example, the fraction of Jensen-Shannon divergence "explained" by a subset of species $\mu \in \mathcal{S}$ can be defined as

$$p(\mathcal{S}) = \frac{\sum_{\mu \in \mathcal{S}} \bar{f}_{\mu} \left[ \left( \frac{2}{1+r_{\mu}} \right) \log_2 \left( \frac{2}{1+r_{\mu}} \right) + \left( \frac{2}{1+r_{\mu}^{-1}} \right) \log_2 \left( \frac{2}{1+r_{\mu}^{-1}} \right) \right]}{JSD}. \quad (9)$$

Figure 2e shows the percent JSD explained by the subset of strains with a minimum fold change $r$, by defining

$$\mathcal{S}(r) = \left\{ \mu : |\log r_{\mu}| \geq |\log r| \right\}. \quad (10)$$

Similarly, Fig. 3 shows the percent JSD explained by a single focal species (Fig. 3c), the entire set of focal species (Fig. 3e), or all species in the same family as one of the focal species (Fig. 3f).

In the latter two cases, we compared these distributions to a null model in which the same number of replacement or modification events were redrawn from the same subset of hosts, by weighting each resident population by the empirical rate of replacement or modification events in that species across our entire dataset (Fig. 3e, f). We assessed the significance of these differences using the mean of this distribution as a test statistic.

## Reporting summary

Further information on research design is available in the Nature Portfolio Reporting Summary linked to this article.

## Data availability

Postprocessed source data for Figs. 1–3 can be found in Supplementary Data 2 and 4; these figures can be regenerated by running the `generate_all_figures.py` script provided in Supplementary Code 1. Raw sequencing data from the Human Microbiome Project are publicly available at the NCBI Sequence Read Archive using the accessions provided in Supplementary Data 1.

## Code availability

All analysis code and figure generation scripts for Figs. 1–3 are available on Github (https://github.com/bgoodlab/microbiome_ecoevo_correlations), as well as in Supplementary Code 1. Figs. 1–3 can be regenerated by running the `generate_all_figures.py` script in the base directory.

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

## Acknowledgements

We thank Z. Liu for identifying a host mislabeling event in the HMP data, and S. Walton for feedback on the manuscript. This work was supported in part by a Bio-X Undergraduate Research Fellowship (to L.B.R.), the Alfred P. Sloan Foundation grant FG-2021-15708 (to B.H.G.), and NIH NIGMS Grant No. R35GM146949 (to B.H.G.). B.H.G. is a Chan Zuckerberg Biohub – San Francisco Investigator.

## Author contributions

Conceptualization: B.H.G.; theory and methods development: B.H.G. and L.B.R.; analysis: B.H.G. and L.B.R.; writing: B.H.G.

## Competing interests

The authors declare no competing interests.
