## [Peer Review File · Nature Communications]

Reviewers' Comments:

Reviewer #1:

Remarks to the Author:

Summary

I had read this preprint in the past and appreciated the opportunity to give it a deep reading for this review. The authors present a series of intuitive and rigorous analyses of high-quality public data. They recapitulate the qualitative patterns they observed using a consumer-resource model of community dynamics, expanding on prior theoretical investigations of the first author. I particularly appreciate the balance the authors struck between their decidedly non-parametric analyses and mathematical modeling, how those analyses led them to consider a consumer-resource model, as opposed to imposing the model first and then performing the analyses. While this may have partially been driven by the limitations of the data (e.g., sample size), I also appreciate that the authors examined the global nature of the patterns, rather than determining whether the pattern exists for each individual species and using those results to tell species-specific stories. There are undoubtedly species that are of greater interest than others to those in applied fields, but at this stage it seems that there remains the need to identify the typical patterns of microbial eco-evolutionary dynamics in the gut.

Something that makes this paper stand-out from the bulk of research into eco-evo dynamics is that the authors work to make the connection between ecology and evolution concrete, specifically through the use of the lead authors' past modeling efforts using MacArthur's consumer-resource model (Good et al., 2018). Given that eco-evo dynamics and interactions are commonly alluded to in the bulk of microbial community studies but rarely defined, it is inspiring to see the authors provide straightforward definitions that build off a workhorse like the Consumer-Resource model. It is particularly interesting how they were able to recapitulate observed patterns in a minimal model that lacked the mechanisms that are often assumed to be necessary to generate eco-evo dynamics (e.g., cross-feeding).

I have a few comments and requests listed below, some of which are questions for the authors that do not necessarily have to be incorporated in the manuscript. Overall, I think the manuscript constitutes a meaningful contribution to the topic of microbial evolution and ecology in the human gut and think it is ready for publication with a few very small tweaks.

Major comments

Lines 39-40: I appreciated that the authors provided a heuristic explanation of the minimal change in abundance that could potentially be related to eco-evo dynamics.

Line 76: Is the increased signal strength at higher taxonomic ranks reflective of conserved metabolic pathways that are relevant for eco-evo dynamics (i.e., coarse-graining to higher taxonomic ranks removing redundant pathways within a taxon) or is it due to the lower number of observables (i.e., taxa) that are used to perform the test?

Lines 87-89: Is this sentence saying that the typical correlation between community diversity and the probability of a modification for a given phyla is much smaller than the difference in correlations between a given pair of phyla? If not I'm unsure what "difference between phyla" refers to.

Lines 141-143: Is this because Shannon's diversity is considered to be robust to sampling effort, so this holds for distances measures with similar forms such as Jensen-Shannon? Or is there something else?

Fig S2: What is the reason for defining richness this way? Is it just that you want to incorporate a frequency cutoff so you can't simply define the presence of a species as one minus the Kronecker delta for the frequency and zero, so, by using the exponential you can approach zero for f_{μ} above the cutoff and approach one below? Or is there another reason?

Fig. S6: I appreciate the authors' use of time-reversal symmetry to describe the order of eco-evolutionary dynamics in this manuscript and in prior work (Garud et al., 2019).

Lines 207-209: This is an interesting point that brings to mind the concept of global constraints, particularly the lead author's past work on macroscopic epistasis as well as the work of others (Good & Desai, 2015; Reddy & Desai, 2021). While the question is beyond the scope of this paper, could the fitness of an invading mutant be viewed as a global constraint so that if you had estimates of X_{μ} you could predict the most likely distribution of Δf_{μ} in response to an invasion?

Lines 377-380; 395-397: Throughout the manuscript the authors take the time to explain how their permutation schemes relate to a given question while preserving meaningful features of the data, something that is useful for the reader but is often not done in studies that use permutation-based tests.

Eq. S1.2: Can you define r_{0} ?

Eq. S1.7, S1.8: I think I have rederived S1.7 by expanding the exponential, using the approximation $1 / (1 + x) \approx 1 - x$, and plugging in S1.6. However, I am having trouble getting back S1.8. Perhaps the authors could include a few additional steps in their derivation of S1.7 and S1.8?

Eq. S1.6 - S1.8: Why did the asymptotic notation go from big "O" to little "o"?

Eq. S1.25: Comparing these two equations, it looks like $\sum_{i,v} \tilde{\alpha}_{i,v}^{-1} X_v \tilde{\alpha}_{\mu, i} = \tilde{X}_{\mu}$. Is this correct? If so, how?

Minor comments

Lines 85-87: Is it possible to include the correlation values here?

Line 97: "their in" => "in their" ?

Line 135: "correlated each other" => "correlated with each other" ?

Line 163: Is "changes" repeated twice?

Line 208: Is there a typo in "the relative uptake rates the focal species"?

Eq. 5b: Is there a particular reason why log base 2 was used? Just curious since the natural log is typically used to calculate diversity, though it wouldn't change the reported results.

Lines 512 - 516: This is a fascinating result.

Line 549: Is there supposed to be a reference here?

References

Garud, N. R., Good, B. H., Hallatschek, O., & Pollard, K. S. (2019). Evolutionary dynamics of bacteria

in the gut microbiome within and across hosts. *PLOS Biology*, 17(1), e3000102.
<https://doi.org/10.1371/journal.pbio.3000102>

Good, B. H., & Desai, M. M. (2015). The Impact of Macroscopic Epistasis on Long-Term Evolutionary Dynamics. *Genetics*, 199(1), 177–190. <https://doi.org/10.1534/genetics.114.172460>

Good, B. H., Martis, S., & Hallatschek, O. (2018). Adaptation limits ecological diversification and promotes ecological tinkering during the competition for substitutable resources. *Proceedings of the National Academy of Sciences*, 115(44), E10407–E10416. <https://doi.org/10.1073/pnas.1807530115>

Reddy, G., & Desai, M. M. (2021). Global epistasis emerges from a generic model of a complex trait. *ELife*, 10, e64740. <https://doi.org/10.7554/eLife.64740>

Reviewer #2:

Remarks to the Author:

In this manuscript the authors analyze data on fecal metagenomes from the HMP to study feedbacks between short-term evolution and ecological structure in the context of the gut microbiome. To do this they identify species that accumulated SNVs across timepoints and further classify these events as evolutionary modifications or strain replacements depending on the total number of accumulated SNVs. They find that the rates at which these events occur are mainly determined by the species identity and not by ecological properties of the community like species diversity. They also find that communities where these events occurred were more likely to undergo taxonomic shifts where not only the evolving species but also distantly related species changed in frequency. Overall, I find that this is a solid study that makes interesting observations about the link between ecological and evolutionary processes in the gut microbiome but I would have appreciated that the authors would discuss in more depth what kind of processes could lead up to the patterns that they observe. The authors do present a model to explain some of their observations but most details about it are in the SI so it's hard to get an intuitive understanding of how the model explains interesting observations like evolving species going down in frequency in some communities. Also, previous studies (some also looking at the gut microbiome) have found a stronger link between the rate of evolutionary change and community properties like species diversity (see Madi et al 2020; Madi et al 2023) which could potentially indicate the presence of ecology-driven feedbacks. It'd be great if the authors could discuss why they think their findings are different and to what extent this could be associated to features of the dataset used (e.g. time between samples) or potential issues with the analysis (e.g. read mismatching). Some other comments below:

Line 49. It'd be great if the authors could also specify in which genes where the SNVs found. Are they similar across related species? Could this information be useful to understand differences like the ones observed between Bacteroidetes and Firmicutes? Also I had a brief look at table S2 and according to the SNVs>100 threshold there are 22 replacement events. Did the authors discard some of the events to get to the number of 16 and what was the criteria?

Line 78. In general the rates of modification are higher than the ones of replacement across all species. Is this because of the distance between timepoints? How do the authors explain this? could the SNV approach be biased towards identifying modification events over replacement ones?

Line 90. It's a bit unclear to me how this graph was made. Aren't there communities where both replacement and modification events happened? If so, are those not plotted here? Also it'd be good to see plot 1C for the probability of a strain replacement event.

Line 137. Why would the power of detection be higher for strain replacement events than modification events?

Line 142. Are the species that went extinct closely related to the evolving species? or do you also find that these species are very distantly related species in the community?

Line 145. I'd change the order of panels a and b to have the latter on top

Line 178. This should be figure 3F

Line 180. I wonder if you could plot the JSD fraction contributed by a species as a function of its phylogenetic distance to the evolving species. Also I think it'd be great if the authors could calculate an index of metabolic similarity between species based on genome annotations and see whether the distantly related species that change in frequency are more (or less) similar to the evolving species than the rest of the community.

Line 188. Not sure how figure 2D illustrates this. Judging from Fig 3A is not clear that there is bias towards focal species expansion at least for strain replacement events.

Line 199. It is not clear to me why the invasion fitness of a mutation would increase if the mutation leads to an increment in the abundance/resource uptake rate of other species (or in other words why is natural selection acting at the community-level in the model). The authors should offer an intuition about this derivation (since this is how they explain why sometimes evolving species go down in frequency) and in general explain the model in more detail. Also I don't understand the statement in line 213 about abundance fluctuations not depending on phenotypic similarity. Isn't the basis of the model that species share a common resource pool (which means that the abundances of two species are more likely to be coupled if they have similar resource utilisation profiles)?

Reviewer #3:

Remarks to the Author:

This is a concise, clearly written paper describing evidence of eco-evolutionary dynamics in the human gut microbiota. The approach and findings are nicely explained and easy to follow, and the work highlights the potential importance of these dynamics in understanding how microbial communities change over time. Given these eco-evolutionary processes are not currently incorporated into many studies of host-associated microbial communities, this seems to be an important contribution to the literature. My comments overall are minor.

Introduction: The relevant material is presented in this section in general. However, I think it would be useful for the authors to comment on eco-evolutionary dynamics more broadly in microbial systems and/or macro-ecosystems. As written, the introduction appeals to a narrow audience of people interested in the gut microbiota, and linking to eco-evolutionary dynamics in other contexts would potentially broaden readership.

Line 77: Given how commonly these phyla are invoked in papers examining variation in the microbiome associated with variation in host environments (and in addressing broad questions such as the importance of diet vs phylogeny), it would be interesting to comment on how modification and replacement rates in these phyla could contribute to some of these patterns. I realize this would have to be somewhat broad and/or speculative, but I think it is worth integrating these ideas.

Line 85: Explain why briefly.

Line 224: I would love a couple examples of more specific applications/implications. It would also be interesting if the authors included some thoughts about whether the relative importance of these eco-evolutionary dynamics is likely to vary by host environment/context and/or host species.

Methods: On the one hand, the authors clearly explain their methods, and I think the overall approach makes sense. On the other hand, using only two timepoints to assess evolution seems like it could

lead to oversimplification or could make it more difficult to understand the finer scale evolutionary dynamics of the system. It would be nice to see the authors reflect on this, even if it is briefly.

Along similar lines, the limitations listed in the supplemental material seem like they would be useful to include in the main text.

Detailed Response to Reviewer Comments

Note: line numbers refer to the track-changes version of the manuscript.

Reviewer #1 (Remarks to the Author):

Summary

I had read this preprint in the past and appreciated the opportunity to give it a deep reading for this review. The authors present a series of intuitive and rigorous analyses of high-quality public data. They recapitulate the qualitative patterns they observed using a consumer-resource model of community dynamics, expanding on prior theoretical investigations of the first author. I particularly appreciate the balance the authors struck between their decidedly non-parametric analyses and mathematical modeling, how those analyses led them to consider a consumer-resource model, as opposed to imposing the model first and then performing the analyses. While this may have partially been driven by the limitations of the data (e.g., sample size), I also appreciate that the authors examined the global nature of the patterns, rather than determining whether the pattern exists for each individual species and using those results to tell species-specific stories. There are undoubtedly species that are of greater interest than others to those in applied fields, but at this stage it seems that there remains the need to identify the typical patterns of microbial eco-evolutionary dynamics in the gut.

Something that makes this paper stand-out from the bulk of research into eco-evo dynamics is that the authors work to make the connection between ecology and evolution concrete, specifically through the use of the lead authors' past modeling efforts using MacArthur's consumer-resource model (Good et al., 2018). Given that eco-evo dynamics and interactions are commonly alluded to in the bulk of microbial community studies but rarely defined, it is inspiring to see the authors provide straightforward definitions that build off a workhorse like the Consumer-Resource model. It is particularly interesting how they were able to recapitulate observed patterns in a minimal model that lacked the mechanisms that are often assumed to be necessary to generate eco-evo dynamics (e.g., cross-feeding).

I have a few comments and requests listed below, some of which are questions for the authors that do not necessarily have to be incorporated in the manuscript. Overall, I think the manuscript constitutes a meaningful contribution to the topic of microbial evolution and ecology in the human gut and think it is ready for publication with a few very small tweaks.

We thank the reviewer for their positive assessment of the manuscript. We have made several modifications to address their specific comments below.

Major comments

(1) Lines 39-40: I appreciated that the authors provided a heuristic explanation of the minimal change in abundance that could potentially be related to eco-evo dynamics.

We are glad to hear that this was helpful – this baseline noise is an inherent feature of natural ecosystems like the gut microbiota, so it was important to figure out a way to control for it.

(2) Line 76: Is the increased signal strength at higher taxonomic ranks reflective of conserved metabolic pathways that are relevant for eco-evo dynamics (i.e., coarse-graining to higher taxonomic ranks removing redundant pathways within a taxon) or is it due to the lower number of observables (i.e., taxa) that are used to perform the test?

That's an interesting question – given the sample sizes involved, we think it is most likely due to the second factor: the error bars at the species level are currently so big a lot of the signal is lost in the sampling noise. It will be interesting to test this hypothesis more systematically as larger species-level samples become available.

(3) Lines 87-89: Is this sentence saying that the typical correlation between community diversity and the probability of a modification for a given phyla is much smaller than the difference in correlations between a given pair of phyla? If not I'm unsure what "difference between phyla" refers to.

We apologize for the lack of clarity – the sentence meant to say that the effect size associated with the Shannon diversity was much smaller than the difference in the average modification rate between phyla. We have rephrased the sentence to make this more clear. (See also our response to Point #13 below)

(4) Lines 141-143: Is this because Shannon's diversity is considered to be robust to sampling effort, so this holds for distances measures with similar forms such as Jensen-Shannon? Or is there something else?

We're not sure what the ultimate cause is, but we mainly trying to show that our initial test wasn't picking up some idiosyncratic feature of the Jensen-Shannon metric. We have rephrased the sentence slightly to make this point more clear.

(5) Fig S2: What is the reason for defining richness this way? Is it just that you want to incorporate a frequency cutoff so you can't simply define the presence of a species as one minus the Kronecker delta for the frequency and zero, so, by using the exponential you can approach zero for f_{μ} above the cutoff and approach one below? Or is there another reason?

Thanks for bringing this up – this soft constraint is similar to a more traditional frequency cutoff, but is slightly more robust to log-scale fluctuations near the minimum frequency threshold (which are common in gut microbiome datasets). We have added a sentence to the caption of Fig. S2 to explain the rationale behind this metric in more detail.

(6) Fig. S6: I appreciate the authors' use of time-reversal symmetry to describe the order of evolutionary dynamics in this manuscript and in prior work (Garud et al., 2019).

Thanks – we think this is an important signal that we hope to follow up on in future work.

(7) Lines 207-209: This is an interesting point that brings to mind the concept of global constraints, particularly the lead author's past work on macroscopic epistasis as well as the work of others (Good & Desai, 2015; Reddy & Desai, 2021). While the question is beyond the scope of this paper, could the fitness of an invading mutant be viewed as a global constraint so that if you had estimates of X_{μ} you could predict the most likely distribution of Δf_{μ} in response to an invasion?

This is an interesting idea. We think that estimating the most likely abundance perturbations will require knowledge about the genetic architecture of the resource uptake phenotypes – this is described in more detail in a new paragraph in the Discussion (lines 268-277).

Nevertheless, if we make some specific assumptions about this genetic architecture, we can make some statistical predictions about the distribution of likely ecological perturbations. We are currently exploring this idea in a follow-up theoretical study using replica-theoretic techniques from statistical physics (see also our response to Comment #10 from Reviewer 2 below).

(8) Lines 377-380; 395-397: Throughout the manuscript the authors take the time to explain how their permutation schemes relate to a given question while preserving meaningful features of the data, something that is useful for the reader but is often not done in studies that use permutation-based tests.

We are glad to hear that this was helpful – we believe that similar permutation tests should have broader utility within microbial ecology, but maintaining the right correlations is crucial.

(9) Eq. S1.2: Can you define r_0 ?

We apologize for not defining this quantity explicitly – the r_0 variable was intended as an arbitrary scale to ensure that the argument of the logarithm in Eq. S1.2 is unitless. Since this is a subtle point (and not particularly critical for the present manuscript) we have removed the r_0 variable in the revised version.

(10) Eq. S1.7, S1.8: I think I have rederived S1.7 by expanding the exponential, using the approximation $1 / (1 + x) \approx 1 - x$, and plugging in S1.6. However, I am having trouble getting back S1.8. Perhaps the authors could include a few additional steps in their derivation of S1.7 and S1.8?

Thanks for bringing this up – reviewing this step we noticed that there was a typo in Eq. S1.8 (the X_{μ} should have been an X_{ν}), so we have corrected this in the revised manuscript.

Eq.S1.8 should directly follow from the definitions in Eqs. S1.9 and S1.10, using the fact that the beta vector sums to one.

(11) Eq. S1.6 - S1.8: Why did the asymptotic notation go from big “O” to little “o”?

Thanks for flagging this. The little “o” notation was left over from a previous version – we’ve changed these all to big “O” in the revised version.

(12) Eq. S1.25: Comparing these two equations, it looks like $\sum_{i,v} \tilde{\alpha}_{i,v}^{-1} X_{v} \tilde{\alpha}_{\mu, i} = \tilde{X}_{\mu}$. Is this correct? If so, how?

Yes this is correct – this fact follows from the definition of the inverse matrix. We’ve added a line explaining this in the revised manuscript.

Minor comments

(13) Lines 85-87: Is it possible to include the correlation values here?

We’re assessing this correlation using logistic regression, so there is not really an analogue of the regression coefficient. However, we agree that providing some information about the associated effect size would be helpful. We’ve added a new Supplemental Table (Table S3) that lists the regression coefficients and p-values for all the logistic regressions we performed in this section.

(14) Line 97: “their in” => “in their” ?

Thanks for catching this typo – we have fixed it in the revised manuscript.

(15) Line 135: “correlated each other” => “correlated with each other” ?

Thanks for catching this typo – we have fixed it in the revised manuscript.

(16) Line 163: Is “changes” repeated twice?

Yes, thanks for catching this typo – we have fixed it in the revised manuscript.

(17) Line 208: Is there a typo in “the relative uptake rates the focal species”?

Yes, thanks for catching this typo – we have fixed it in the revised manuscript.

(18) Eq. 5b: Is there a particular reason why log base 2 was used? Just curious since the natural log is typically used to calculate diversity, though it wouldn’t change the reported results.

Yes, we used base 2 for all the information theoretic quantities because this choice ensures that

the maximum possible Jensen-Shannon divergence is 1.

(19) Lines 512 – 516: This is a fascinating result.

Thanks – we also find it really curious that there’s a “phase transition” here depending on how much “pure fitness” differences contribute to the steady-state abundances.

(20) Line 549: Is there supposed to be a reference here?

Yes, thanks for catching this – we’ve fixed this in the revised manuscript. (This particular paragraph was moved into the main text, and the offending sentence was rephrased so that the reference comes in a later paragraph.)

Reviewer #2 (Remarks to the Author):

In this manuscript the authors analyze data on fecal metagenomes from the HMP to study feedbacks between short-term evolution and ecological structure in the context of the gut microbiome. To do this they identify species that accumulated SNVs across timepoints and further classify these events as evolutionary modifications or strain replacements depending on the total number of accumulated SNVs. They find that the rates at which these events occur are mainly determined by the species identity and not by ecological properties of the community like species diversity. They also find that communities where these events occurred were more likely to undergo taxonomic shifts where not only the evolving species but also distantly related species changed in frequency.

(1) Overall, I find that this is a solid study that makes interesting observations about the link between ecological and evolutionary processes in the gut microbiome but I would have appreciated that the authors would discuss in more depth what kind of processes could lead up to the patterns that they observe. The authors do present a model to explain some of their observations but most details about it are in the SI so it’s hard to get an intuitive understanding of how the model explains interesting observations like evolving species going down in frequency in some communities.

We thank the reviewer for their positive assessment of our manuscript. We appreciate the suggestion to expand our discussion of the underlying mechanisms that could lead to the patterns we observe. To address this comment, we have greatly expanded our discussion of the modeling section to bring in more of the details that were previously left in the SI. In particular, we have added a new figure to the main text (Fig. 4) to provide some more intuition for how the model explains some of the more interesting observations (e.g. evolving species declining in abundance). See also our response to the Reviewer’s comment #10 below.

(2) Also, previous studies (some also looking at the gut microbiome) have found a stronger link between the rate of evolutionary change and community properties like species diversity (see Madi et al 2020; Madi et al 2023) which could potentially indicate the presence of ecology-

driven feedbacks. It'd be great if the authors could discuss why they think their findings are different and to what extent this could be associated to features of the dataset used (e.g. time between samples) or potential issues with the analysis (e.g. read mismapping).

Thanks for bringing this up – the Madi et al 2023 paper came out after our study was preprinted, so we have not yet had a chance to integrate their results into our paper. Figs. 4 and 5 of the Madi et al 2023 study asks a similar question that we are asking in our Fig. 1C, using slightly different input and output variables (see below). However, we would disagree with the conclusion that Madi et al found a stronger link between the rate of evolutionary change and community properties like species diversity. We believe that their analysis suffers from a few methodological issues, which, when properly accounted for, bring their results into accordance with what we've found in Fig. 1C. We first describe these issues below, and conclude by discussing the revisions we have made to address the Reviewer's comment.

Methodological issues in Madi et al 2023. The biggest difference is that the analysis in Madi et al 2023 does not explicitly distinguish between (i) evolutionary modification events, (ii) strain replacements events, (iii) and the more general phenomenon of two co-colonizing strains exhibiting small fluctuations in abundance over time ("strain fluctuations"). Instead, Madi et al treated their various outputs (e.g. # of gene gains or losses, or the change in levels of within-host polymorphism over time) as continuous variables, and looked for statistically significant associations with community diversity at the initial timepoint. This methodological choice has important implications for the downstream interpretation of their results.

For example, from the magnitudes of the genetic changes that Madi et al plotted in Figs. 4 and 5, one can infer that their signal must be dominated by strain replacement events and/or strain fluctuations – these numbers are too high to be consistent with previous observations of de novo evolutionary changes within hosts (e.g. Garud & Good et al, *PLoS Bio* 2019; Zhao & Lieberman et al, *Cell Host & Microbe* 2019). However, once we know that signal is dominated by strain replacements and/or strain fluctuations, then ***we can conclude that the regression coefficients inferred by Madi et al do not actually quantify the rates of the underlying events*** (which we quantify in Fig. 1C and our new Table S3 and Fig. S3) but rather the *product* of the rate and the magnitude of the changes associated with it (the vast majority of which, by definition, accumulated before the strains co-colonized the host in question). The strain fluctuation scenario involves a further confounder, which is the prior probability that multiple co-colonizing strains are present in the same host in the first place.

We suspect that this latter effect is likely driving the "evolutionary rate" correlations reported by Madi et al. In the first part of their paper, Madi et al showed that communities with higher Shannon diversity are more likely to have multiple co-colonizing strains of the same species (Figs. 2 & 3). Thus, higher diversity communities should also be more likely to exhibit signatures of strain fluctuations (e.g. changing polymorphism levels or differences in gene content) even if the *conditional* probability of strain fluctuations remains constant. The regressions in Figs. 4 & 5 of Madi et al do not control for these differences in co-colonization, so we suspect that they are simply recapturing the same colonization correlations that were reported in Figs. 2 & 3. This

hypothesis is supported by the fact that many of the species with significant correlations in Madi et al's Fig. 2 also tend to have significant correlations in their Fig. 4, and vice versa, as expected if they were capturing similar effects.

In contrast, our quasi-phasing approach allows us to distinguish between these confounding factors, and thereby measure a true correlation between community diversity and the rates of selective sweeps. We observe no correlation between the initial community diversity and the rate of strain replacement (now illustrated in Table S3 and Figure S3; see our response to the Reviewer's point #5 below). This result is consistent with our interpretation that Madi et al's results are largely driven by strain fluctuations, rather than differences in evolutionary rates.

Changes to revised manuscript. We recognize that these are subtle points, and we do not wish for our manuscript to become a "take down" of the Madi et al study. As a compromise, we have revised the manuscript to expand our discussion of the results in Fig. 1C, and provided a citation to Madi et al (2023) for performing a related analysis (lines 102-103). We have also added a few citations for works addressing similar questions in non-gut systems (see our response to comment #1 from Reviewer #3 below). Together, these changes should help readers evaluate our findings in the context of this earlier literature. We believe that our statistical framework for disentangling rates and magnitudes of evolutionary events will be useful for addressing this same question in other microbial ecosystems.

Some other comments below:

(3) Line 49. It'd be great if the authors could also specify in which genes where the SNVs found. Are they similar across related species? Could this information be useful to understand differences like the ones observed between Bacteroidetes and Firmicutes? Also I had a brief look at table S2 and according to the SNVs>100 threshold there are 22 replacement events. Did the authors discard some of the events to get to the number of 16 and what was the criteria?

Thank you for catching this – this discrepancy made us realize that we forgot to explain one additional filtering step. In analyzing some of this same data for a different paper, one of the other graduate students in the lab discovered that the follow-up samples from two subjects (sample ids 763536994 and 763880905) were possibly swapped in the HMP dataset. We noticed this because both subjects exhibited strain replacement events in all of their resident species (and were therefore outliers in Fig. 1D), but they had nearly identical strains when compared to the corresponding sample from the other host (which is never observed among other between-host comparisons). Given this observation, we had omitted these two subjects from all of our downstream analyses so that they would not bias our results. However, they were still included in Table S2 for completeness.

We had previously mentioned this issue in our acknowledgements section, but had forgotten to explain it in the methods. We have now added some sentences explaining this issue (lines 416-421).

(4) Line 78. In general the rates of modification are higher than the ones of replacement across all species. Is this because of the distance between timepoints? How do the authors explain this? could the SNV approach be biased towards identifying modification events over replacement ones?

Great observation – this is something we analyzed in detail in our previous paper (Ref 4). In that work, we showed that on long sufficiently timescales (e.g. >10-20 yrs), most species experience at least one strain replacement event (which wipes out the record of any previous modification). But on the shorter timescales of the HMP cohort, evolutionary modification events are >5x more common than replacements. It wasn't obvious *a priori* that this had to be the case – we think it's telling us something interesting about how easily a selection pressure is "filled" by a new mutation event vs the invasion of some pre-existing strain.

What we can say rather confidently is that these differences are not caused by a bias toward identifying modification events over replacements. The biases in our approach tend to run in the other direction: it's much easier for us to call a strain replacement event than it is to call an evolutionary modification (we describe the reasons for this in our response to the Reviewer's point #6 below). If anything, we think it is more likely that we are missing a large fraction of evolutionary modification events (which could partially explain some of the spread in the "none" distribution in Fig. 2D).

(5) Line 90. It's a bit unclear to me how this graph was made. Aren't there communities where both replacement and modification events happened? If so, are those not plotted here?

We apologize for the lack of clarity here. Yes, there were communities that experienced both replacement and modification events – these are included in the right half of Fig 1D. We have rephrased the figure legend to explain this better.

Also it'd be good to see plot 1C for the probability of a strain replacement event.

Thanks for pointing this out – we had performed this analysis in the previous version of the manuscript, but this comment made us realize that the negative result for strain replacements was only left implicit in the text. Unfortunately, the total number of strain replacement events is too small for an analogous version of Fig. 1C to be an effective visualization (there is not enough data to plot the probability estimates in different quantiles). However, we can still perform the analogous logistic regression, and we find there is no significant correlation in this case. To clarify this point, we have now explicitly referenced this lack of correlation in the legend of Fig. 1C, as well as in the main text (lines 107-108). We have also added a new supplemental figure (Fig. S3) comparing the initial entropies of the strain replacement events with their background distribution. This figure shows that the lack of signal is not just a sample size issue – there is no enrichment that is even apparent by eye.

(6) Line 137. Why would the power of detection be higher for strain replacement events than modification events?

The reason for this comes down to the numbers of SNVs involved: strains replacement events tend to involve large numbers of SNVs (1000-10000) that are scattered across the genome. It is only necessary to measure a fraction of these (e.g. >10%) in order to confidently detect a strain replacement event. In contrast, an evolutionary modification event may involve just a handful of SNVs. If these happen to lie in regions that we were unable to examine in that host (e.g. a gene that is not present on the reference genome, or in a region we had to filter out in our effort to reduce bioinformatic artifacts), then there is a possibility that we may not detect any SNV differences in that particular species over time. We expect this to lead to a larger false negative rate for evolutionary modifications vs replacements.

To clarify this issue, we have modified this sentence to remind the reader that strain replacements are genome-wide events, which should give some indication of why there might be greater power to detect them.

(7) Line 142. Are the species that went extinct closely related to the evolving species? or do you also find that these species are very distantly related species in the community?

Great question – these also tend to be rather distantly related to the focal species. While we had previously described one of these examples in words (lines 195-198), this comment made us realize that it would be helpful to characterize this phenomenon more systematically. We have therefore added two new panels to Figure 3 (panels g and h) showing the taxonomic relationships of the extinct-vs-focal species. These data show that the extinct species tend to come from distinct microbial families, and do not seem to exhibit any enrichment in taxonomic distance to the focal species.

(8) Line 145. I'd change the order of panels a and b to have the latter on top

Thanks for the suggestion – we have switched these panels in the revised version of Figure 3.

(9) Line 178. This should be figure 3F

Thanks for catching this – we have fixed this in the revised manuscript.

(10) Line 180. I wonder if you could plot the JSD fraction contributed by a species as a function of its phylogenetic distance to the evolving species. Also I think it'd be great if the authors could calculate an index of metabolic similarity between species based on genome annotations and see whether the distantly related species that change in frequency are more (or less) similar to the evolving species than the rest of the community.

Thanks for these suggestions. We found that the joint distribution of JSD fraction vs phylogenetic distance was hard to visualize directly: there is a lot of spread in this distribution, which makes it difficult to compare the data to its null expectation. As a compromise, we made a new supplemental figure (Fig. S9) containing analogous versions of Figs. 3E,F for different

levels of taxonomic divergence (same species, same genus, same family, and same phylum). As in the original plots in Figs. 3E,F, we find that none of these taxonomic levels explain more of the JSD than expected by chance.

Re: metabolic similarity, we are also very interested in this question. However, we think that doing this properly will require better estimates of metabolic similarity than we can currently get with reference genome annotations.

In unpublished theoretical work, we have been able to recapitulate these extinction dynamics using a generalization of the resource competition model in Eq. (1). Interestingly, in this case – where we have complete knowledge of the underlying metabolic phenotypes – we can show that the species that go extinct are roughly as metabolically diverged from the focal species as they are from another random strain. This is plotted in panel C of the figure reproduced below:

Figure 4 from McEnany & Good (in prep)

Figure 4: **Successful mutations lead to extinction events in other niches.** (A) Schematic showing the extinction of an unrelated species (blue) after a beneficial knockout mutation (orange) invades. We can analyze whether the blue species has a similar resource strategy to the orange mutant, or whether it uses the resource targeted by the mutation (resource 1). (B) Average number of extinctions among species in the population (besides the parent species) after mutant invasion, as a function of niche saturation S^*/\mathcal{R} , for two values of \mathcal{R}_0 . Inset: full probability distribution of number of extinctions for the starred point, compared to the Poisson fit matched to the zero-extinction probability. (C) CCDF of the number of metabolized resources shared between the mutant species and the non-parent species it drives to extinction, again for the starred point in (A). Red curves show the analogous background distribution of number of shared resources between the mutant and *all* species in the population, regardless of whether they become extinct. (D) Probability of extinction as a function of initial relative abundance in the population, for the starred point in (A). (E) The fold change in probability that a species driven to extinction uses the same resource targeted by the mutant (i.e., the resource being knocked in or out), relative to the background distribution of resource use in the population. Data is shown for knock-in and knock-out mutations at two values of niche saturation, as a function of \mathcal{R}_0 . All simulations were performed with 10,000 trials at $N = 500$ and $S^*/S = 0.1$; error bars show counting error.

In place of this enrichment in overall metabolic similarity, we find that the extinct species are somewhat more likely to use (or not use) the specific resource targeted by the successful mutation (depending on whether it is a gain of function or loss of function mutation). This is illustrated in panel E above.

We are excited to test this theoretical prediction, perhaps using some of the observations compiled in the present study. However, the subtle nature of the signal suggests that this will require rather fine-grained estimates of the metabolic phenotypes of the different species.

Moreover, our theory suggests that it is not sufficient to simply compare the metabolic potential of the entire genome, but rather the realized metabolic niche of each species within the in vivo environment. We are currently trying to this using genome-scale metabolic models, which provide a prediction for this realized metabolic niche. However, this analysis has proven to be sufficiently complex that we think that it is best left for a future study.

At a coarser level, we think that much of this metabolic information should be contained in the new taxonomic decomposition in Fig. S9. Extensions of this approach to more fine-grained metabolic phenotypes will be an interesting avenue for future work.

Line 188. Not sure how figure 2D illustrates this. Judging from Fig 3A is not clear that there is bias towards focal species expansion at least for strain replacement events.

We apologize for the confusion – this sentence was supposed to be referencing a result from a previous paragraph that showed a slight enrichment for positive frequency changes for species that experienced a strain replacement event. We have rephrased the sentence to more clearly indicate that it is talking about evolutionary modifications specifically, and added an additional reference to panel A, where this signal is potentially more evident.

Line 199. It is not clear to me why the invasion fitness of a mutation would increase if the mutation leads to an increment in the abundance/resource uptake rate of other species (or in other words why is natural selection acting at the community-level in the model). The authors should offer an intuition about this derivation (since this is how they explain why sometimes evolving species go down in frequency) and in general explain the model in more detail.

Yes, we agree that this is a very interesting mathematical result, since it seems to run so counter to our normal intuition. To better explain this phenomenon, we have greatly expanded the modeling section in the revised manuscript. In particular, we have added a new main figure (Fig. 4) which both explains the setup of the model in greater detail (Fig. 4A,B), and provides a concrete example illustrating how a beneficial mutation can lower the abundance of its focal species (Fig. 4C). We have also added an additional paragraph discussing the limitations of this analysis (see our response to point #7 from Reviewer 3 below). We hope these changes will provide better intuition for what is going on here.

Also I don't understand the statement in line 213 about abundance fluctuations not depending on phenotypic similarity. Isn't the basis of the model that species share a common resource

pool (which means that the abundances of two species are more likely to be coupled if they have similar resource utilisation profiles)?

Thanks for bringing this up – this is a subtle but important point. It is true that two species are more likely to be coupled if they have similar resource utilization profiles. However, our result in Eq. 1 shows that as long as they differ in some way (which is necessary in order for them to stably coexist), then it is possible to find *some* mutation that decouples their abundance fluctuations. The likelihood of these mutations could still be small – answering this question would require additional assumptions about the genetic architecture of the resource utilization rates. We now discuss this issue in more detail in the new limitations paragraph on lines 264-277 (see our response to point #7 from Reviewer 3 below). Together with the new information presented in Fig. 4, this should hopefully address the Reviewer’s question.

Reviewer #3 (Remarks to the Author):

This is a concise, clearly written paper describing evidence of eco-evolutionary dynamics in the human gut microbiota. The approach and findings are nicely explained and easy to follow, and the work highlights the potential importance of these dynamics in understanding how microbial communities change over time. Given these eco-evolutionary processes are not currently incorporated into many studies of host-associated microbial communities, this seems to be an important contribution to the literature. My comments overall are minor.

We thank the reviewer for their positive words about the manuscript.

(1) Introduction: The relevant material is presented in this section in general. However, I think it would be useful for the authors to comment on eco-evolutionary dynamics more broadly in microbial systems and/or macro-ecosystems. As written, the introduction appeals to a narrow audience of people interested in the gut microbiota, and linking to eco-evolutionary dynamics in other contexts would potentially broaden readership.

Thanks for pointing this out. In the revised manuscript, we have expanded the second paragraph of the introduction (lines 28-33) to more explicitly highlight the links other microbial systems and macro-ecosystems. We have also added some additional background and citations to other systems when discussing the correlations between community diversity and the rate of evolution in the Results section (lines 93-103; see our response to the Reviewer’s comment #3 below). Finally, we have expanded our discussion of the extension to other microbial ecosystem in the concluding paragraph of the manuscript (lines 294-298). While our work is still focused on the gut microbiota as a model system, these changes should hopefully highlight the many conceptual connections to the broader literature on evo-evolutionary dynamics.

(2) Line 77: Given how commonly these phyla are invoked in papers examining variation in the microbiome associated with variation in host environments (and in addressing broad questions such as the importance of diet vs phylogeny), it would be interesting to comment on how modification and replacement rates in these phyla could contribute to some of these patterns. I

realize this would have to be somewhat broad and/or speculative, but I think it is worth integrating these ideas.

Thanks for pointing this out. We hadn't thought about this connection before, but agree that it might be worth a brief reference. We have added a new sentence to the discussion along these lines (lines 289-293). We have also added a new supplemental figure (Fig. S4) that shows that the Firmicutes-to-Bacteroidetes ratio does not provide any additional explanatory power for the rate of within-host evolution, beyond that expected from the phylum of the focal species.

(3) Line 85: Explain why briefly.

Thanks for the suggestion – we have expanded discussion of these classical hypotheses, and have included some citations of recent experimental work on this question in non-gut settings (this should also help address the Reviewer's comment #1 above).

(4) Line 224: I would love a couple examples of more specific applications/implications. It would also be interesting if the authors included some thoughts about whether the relative importance of these eco-evolutionary dynamics is likely to vary by host environment/context and/or host species.

Thanks for the suggestion – we have added some concrete examples of possible implications to this paragraph (lines 285-293), including the Firmicutes:Bacteroidetes point above.

(6) Methods: On the one hand, the authors clearly explain their methods, and I think the overall approach makes sense. On the other hand, using only two timepoints to assess evolution seems like it could lead to oversimplification or could make it more difficult to understand the finer scale evolutionary dynamics of the system. It would be nice to see the authors reflect on this, even if it is briefly.

We agree – we have added a brief reference to this limitation in the concluding paragraph (lines 296-297).

(7) Along similar lines, the limitations listed in the supplemental material seem like they would be useful to include in the main text.

We agree – we have now moved this entire paragraph to the main text (lines 264-267).

Reviewers' Comments:

Reviewer #1:

Remarks to the Author:

The authors have addressed all my comments and have gone to a considerable effort to strengthen an already solid manuscript. Figure 4 is a particularly nice addition. I have no additional comments and believe that the manuscript should be published.

Reviewer #2:

Remarks to the Author:

I thank the authors for addressing the concerns I raised and discussing thoroughly the differences between their work and previous studies. Some final comments:

* Thanks for providing a figure to illustrate the model. However I'm not sure I understand why the dotted line is only determined by β_1 and not by the influx of resource 1 relative to the one of resource 2 (β_2). The optimal strategy would be to match α to that ratio, no? In general a little bit more text describing the figure would be useful.

*Line 252. I now get that these mutations are beneficial because otherwise the strains would go extinct. So even though they lead to a reduction in frequency the alternative would be that the strain is outcompeted by the strains with higher total uptake rate. I think it'd be useful to point this out in the text.

*Not sure about the quality of the annotations that can be made and whether the authors already looked into this but as I had mentioned previously it'd be great if the authors could include information on the identity of genes where these mutations are occurring.

*Line 32. Preposition missing

Reviewer #3:

Remarks to the Author:

My comments have all been addressed satisfactorily.

Detailed Response to Reviewer Comments

Reviewer #1 (Remarks to the Author):

The authors have addressed all my comments and have gone to a considerable effort to strengthen an already solid manuscript. Figure 4 is a particularly nice addition. I have no additional comments and believe that the manuscript should be published.

We are glad to hear that our revisions have addressed the reviewer's comments.

Reviewer #2 (Remarks to the Author):

I thank the authors for addressing the concerns I raised and discussing thoroughly the differences between their work and previous studies. Some final comments:

(1) Thanks for providing a figure to illustrate the model. However I'm not sure I understand why the dotted line is only determined by β_1 and not by the influx of resource 1 relative to the one of resource 2 (β_2). The optimal strategy would be to match α to that ratio, no? In general a little bit more text describing the figure would be useful.

Thanks for pointing this out – the reviewer is essentially correct, but our derivation in the Supplementary Information adopted the convention that the influxes are normalized ($\beta_1 + \beta_2 = 1$), so β_1 is also equal to the fractional influx, $\beta_1 / (\beta_1 + \beta_2)$. To eliminate confusion, we've changed the figure to display the fractional influx, $\beta_1 / (\beta_1 + \beta_2)$ instead of β_1 . We have also added some additional text to the legend of Fig. 4C to describe this figure in more detail.

(2) Line 252. I now get that these mutations are beneficial because otherwise the strains would go extinct. So even though they lead to a reduction in frequency the alternative would be that the strain is outcompeted by the strains with higher total uptake rate. I think it'd be useful to point this out in the text.

Interestingly, the reviewer's intuition is not entirely correct – the bottom example in Fig. 4C does not have a higher total uptake rate than the strain it outcompetes (in fact, one can lower the total uptake rate of this example slightly and it will still be favored to invade). This is because the invasion fitness of a mutation depends on both the total uptake rate (X) and the resource preference (α) of the mutant in a somewhat complicated fashion, which depends on the phenotypes of the two resident strains. We've added some text in the legend of Fig. 4C to point this out in more detail.

Unfortunately, we are not aware of any simple explanation for this result, other than the mathematical derivation in Supplementary Notes. We have therefore added a link to the Supplementary Notes in the legend of Fig. 4C so the reader can know where these results come

from.

(3) Not sure about the quality of the annotations that can be made and whether the authors already looked into this but as I had mentioned previously it'd be great if the authors could include information on the identity of genes where these mutations are occurring.

Our apologies for overlooking this request in the last round of revisions. We have now added a new Supplementary Data file (Supplementary Data 3) that provides a complete list of all the SNVs associated with each of the within-host sweeps we analyzed in this study, including the position on the reference genome and the associated gene IDs.

(4) Line 32. Preposition missing

Thanks for flagging this – we have corrected it in the revised manuscript.

Reviewer #3 (Remarks to the Author):

My comments have all been addressed satisfactorily.

We are glad to hear that our revisions have addressed the reviewer's comments.